# Motion Forecasting in Continuous Driving

**Nan Song**[1]    **Bozhou Zhang**[1]    **Xiatian Zhu**[2]    **Li Zhang**[1]*

[1]School of Data Science, Fudan University    [2]University of Surrey

https://github.com/fudan-zvg/RealMotion

## Abstract

Motion forecasting for agents in autonomous driving is highly challenging due to the numerous possibilities for each agent's next action and their complex interactions in space and time. In real applications, motion forecasting takes place repeatedly and continuously as the self-driving car moves. However, existing forecasting methods typically process each driving scene within a certain range *independently*, totally ignoring the situational and contextual relationships between successive driving scenes. This significantly simplifies the forecasting task, making the solutions suboptimal and inefficient to use in practice. To address this fundamental limitation, we propose a novel motion forecasting framework for continuous driving, named **RealMotion**. It comprises two integral streams both *at the scene level*: (1) The scene context stream progressively accumulates historical scene information until the present moment, capturing temporal interactive relationships among scene elements. (2) The agent trajectory stream optimizes current forecasting by sequentially relaying past predictions. Besides, a data reorganization strategy is introduced to narrow the gap between existing benchmarks and real-world applications, consistent with our network. These approaches enable exploiting more broadly the situational and progressive insights of dynamic motion across space and time. Extensive experiments on Argoverse series with different settings demonstrate that our RealMotion achieves state-of-the-art performance, along with the advantage of efficient real-world inference.

## 1   Introduction

Motion forecasting is a crucial element in contemporary autonomous driving systems, enabling self-driving vehicles to predict the movement patterns of surrounding agents [43, 17]. This prediction is vital for ensuring the safety and reliability of driving. However, numerous complex factors, including stochastic road conditions and the diverse motion modes of traffic participants, make resolving this task challenging. Recent developments have focused on the study of representation and modeling [10, 52, 51], in tandem with a growing emphasis on precise trajectory predictions [6, 32, 49, 15, 50, 35]. Furthermore, the field has witnessed an increased focus on multi-agent forecasting, a more challenging yet valuable subtask [26, 1, 14, 31]. These advancements have collectively contributed to substantial progress in motion forecasting in recent years.

However, we realize that existing methods tackle motion forecasting tasks in an isolated manner, i.e., they treat every individual driving scene within a limited range independently, overlooking that in reality motion forecasting is inherently temporally interrelated while any ego-car drives on. That means previous methods ignore the driving context across successive scenes, as well as the corresponding potentially useful information from previous driving periods (Fig. 1).

---

*Li Zhang (lizhangfd@fudan.edu.cn) is the corresponding author.

38th Conference on Neural Information Processing Systems (NeurIPS 2024).

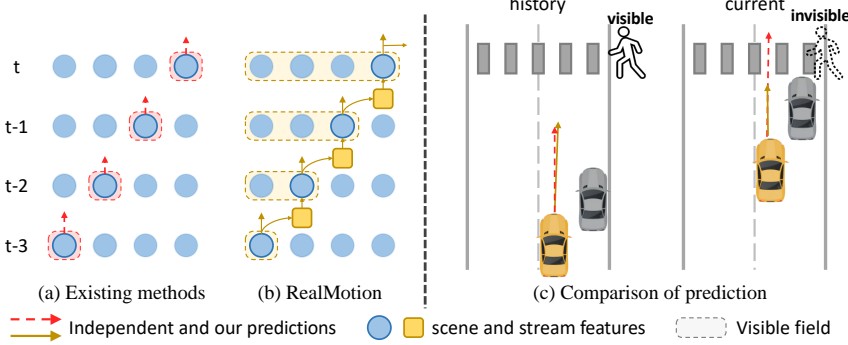

Figure 1: Comparison of **(a) existing methods** independently processing each scene and **(b) our Real-Motion** recurrently collecting historical information. **(c)** For example, RealMotion can perceive the currently invisible pedestrian and predict the giving way for the interested agent.

Under the above insight and consideration, we propose an efficient in-context motion forecasting framework for continuous driving, named **RealMotion**. It comprises two streams to transit states of scenes: (1) A scene context stream that accumulates historical scene context progressively, capturing temporal interactions among scene elements and addressing complex driving situations. (2) An agent trajectory stream that continually optimizes the predictions for dynamic agents like vehicles, considering temporal consistency constraints and capturing precise motion intention. Each stream utilizes a specially designed cross-attention mechanism to transit scene states and fulfill its function.

Our **contributions** are summarized as follows: **(i)** We solve the motion forecasting problem from a perspective of the real-world applications, which enables the extraction and utilization of valuable situational and progressive knowledge. **(ii)** We introduce RealMotion, a novel motion forecasting approach that sequentially leverages both scene context and predicted agent motion status over time, meanwhile maintaining a lower real-world inference latency. **(iii)** To support the continuous driving setting on existing benchmarks, we implement a data reorganization strategy to generate scene sequences, closely simulating the real-world driving scenarios. Extensive experiments on Argoverse series with different settings demonstrate that RealMotion achieves state-of-the-art performance.

## 2  Related work

In autonomous driving, accurately predicting future trajectories of agents of interest relies on an appropriate representation of scene elements. Early methods [29, 12, 2] rasterize driving scenarios into images and utilize off-the-shelf convolutional networks for scene context encoding. However, due to their limited ability to capture intricate structural information, recent studies [49, 15, 52, 36] have shifted towards vectorized representations, as exemplified by the emergence of VectorNet [10]. Additionally, graph-based structures are widely adopted to model dynamics, interactions, and relationships among agents and maps [22, 13, 44, 19, 18, 9].

With the encoded scene features, various approaches are explored for estimating multi-modal future trajectories. Early methods focus on goal-based prediction [49, 15] or use probability distribution heatmaps for trajectory sampling [12, 13]. Recent approaches like HDGT [19], Wayformer [25], and others [23, 26, 48, 32, 51] leverage Transformer architectures [37] to model detailed relationships within the overall scene. Moreover, there are methods introducing novel paradigms (e.g. pre-training [5, 4, 21, 28], post-refinement [6, 50] or Diffusion [20]) to achieve impressive performance.

To address the relevance of predicted trajectories for different agents in real-life scenarios, recent efforts have focused on multi-agent forecasting. Some methods [14, 38, 52, 33] adopt an agent-centric design, iteratively predicting trajectories for each agent, which can be inefficient and hinder exploration of relationships among agents. Conversely, SceneTransformer [26] introduces a scene-centric framework for joint forecasting of all agents, presenting a novel design. Moreover, recent methods explore a query-centric design [51] or consider adaptations [1] to address this task.

Nevertheless, the methods mentioned above independently perform forecasting for each scene sample, which conflicts with practical settings where driving scenarios are interconnected over time. To overcome this limitation, pioneering work [27] first introduce a temporal motion benchmark based

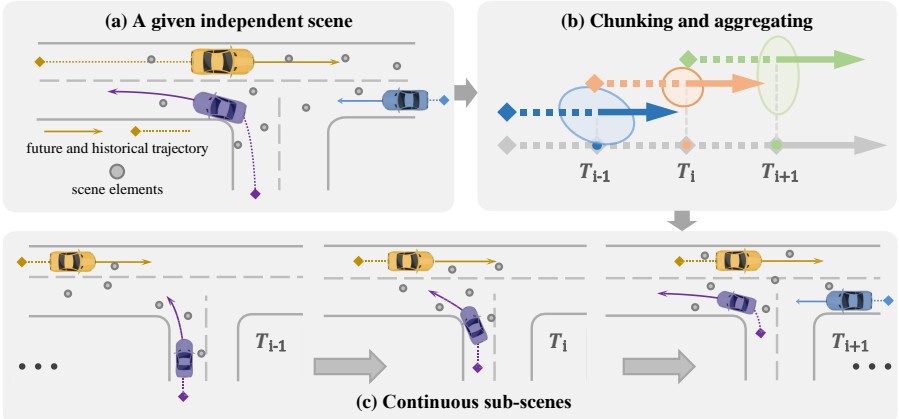

Figure 2: Illustration of our data reorganization strategy, processing (a) a given independent scene by (b) chunking the trajectories into segments and aggregating surrounding elements, generating the (c) continuous sub-scenes.

on tracking dataset. Instead, we design a transformed data structure to mimic real-world conditions and propose RealMotion to effectively model the temporal relationships.

## 3 Methodology

### 3.1 Preliminary

**Data reorganization.**   Considering the discrepancy between existing benchmarks and practical applications, our first step is to reorganize these datasets by transforming each sample scene into a continuous sequence, mimicking the successive real-world driving scenarios. Specifically, we retrospectively examine each independent scene by evenly splitting agent trajectories into shorter segments and sampling local map elements (refer to Fig. 2). Specifically, we first select several split points $T_i$ along historical frame steps. Next, we generate trajectory segments of identical length by extending from these points both into the past and the future. The number of historical and future steps is determined by the minimum split point and the length of ground-truth trajectory, respectively. Also, surrounding agents within a certain range and a local map are aggregated for interested agents at each split point, forming a sequence of sub-scenes. This reorganization allows freely capitalizing on the original elements to offer valuable situational and progressive insights at the scene level for model optimization. Hence, existing methods can also involve and benefit from the novel data structure. We have implemented this approach within the state-of-the-art method QCNet [51], which is further discussed in the appendix, highlighting the generality of our data structure.

**Input representation.**   In the context of motion forecasting, the trajectories of agents and a high-definition road map are provided in the driving scenario. Following [10], we transform these scene elements into vectorized representations as input. The historical trajectories are denoted as $A \in \mathbb{R}^{N_a \times T \times C_a}$, where $N_a$, $T$, and $C_a$ represent the number of agents, the number of historical frames, and the motion states (e.g., position, angle, velocity, and acceleration), respectively. The road map is divided into several lane segments, denoted as $M \in \mathbb{R}^{N_m \times P \times C_m}$, where $N_m$, $P$, and $C_m$ indicate the number of lane segments, the points of each segment, and the lane features (typically represented as coordinates). All these states are normalized relative to the current position of the agent of interest.

### 3.2 RealMotion for continuous forecasting

As depicted in Fig. 3, our ***RealMotion*** approach comprises an encoder, a decoder, a scene context stream, and an agent trajectory stream. Following the encoder-decoder structure, the two streams are designed to execute temporal modeling, focusing on context information and trajectory prediction along the time dimension.

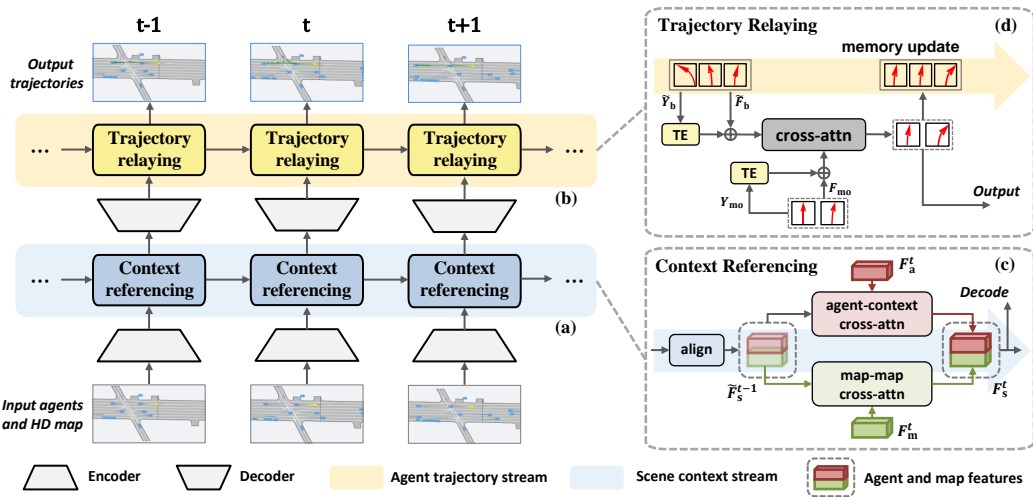

Figure 3: Overview of our RealMotion architecture. RealMotion adopts an encoder-decoder structure with two intermediate streams designed to capture interactive relationships within each scene and across the continuous scenes. The (a) **Scene context stream** and (b) **Agent trajectory stream** iteratively accumulate information for the scene context and rectify the prediction, respectively. The (c) **context referencing** and (d) **trajectory relaying** modules are specially-designed cross-attention mechanism for each stream.

### 3.2.1 Multimodal feature encoding and trajectory decoding

For scalability and simplicity, we adopt the plain encoder and decoder design following [5, 32]. Specifically, we encode the map features $F_\mathrm{m} \in \mathbb{R}^{N_\mathrm{m} \times D}$ by a PointNet-like encoder [30], where $D$ refers to the embedding dimension. As [5], we extract the agent features $F_\mathrm{a} \in \mathbb{R}^{N_\mathrm{a} \times D}$ by stacked neighborhood attention blocks [16]. Given the agent and map features, we concatenate them to derive the scene features $F_\mathrm{s} \in \mathbb{R}^{(N_\mathrm{a}+N_\mathrm{m}) \times D}$. We then employ a Transformer [37] encoder to learn the interrelationships of these features.

For trajectory prediction along with the probability, we utilize two Multilayer Perceptron (MLP) modules. Additionally, we forecast a singular trajectory for auxiliary training purposes, focusing on the movement patterns of other agents.

### 3.2.2 Scene context stream

The scene context stream is meticulously designed to progressively gather information about the surrounding environment, thereby enhancing trajectory prediction. Positioned after the encoder, this component plays a crucial role as the historical scene profoundly influences the understanding of temporal motion behaviors exhibited by agents. For instance, the ability to estimate the trajectory in complex scenes or evaluate currently occluded agents can be greatly improved by taking into account the previous situation and context.

At the current time $t$, we denote the current and historical scene features as $F_\mathrm{s}^t$ and $F_\mathrm{s}^{t-1}$, respectively, extracted from the respective local coordinate systems. The process begins by projecting $F_\mathrm{s}^{t-1}$ onto the current system. To achieve this, the two features must be aligned, considering the gap of motion between them. Motion-aware Layer Normalization (MLN) [40] is employed for this purpose:

$$\tilde{F}_\mathrm{s}^{t-1} = \mathrm{MLN}(F_\mathrm{s}^{t-1}, \ \mathrm{PE}([\Delta x, \Delta y, \Delta \theta, \Delta t]), \tag{1}$$

where $\Delta \theta$ denotes the heading angles, $\Delta t$ refers to the timestamps, and $(\Delta x, \Delta y)$ represents the difference between their positions. PE indicates the position encoding function. Subsequently, we repartition the scene features $\tilde{F}_\mathrm{s}^{t-1}$ and $F_\mathrm{s}^t$ into the agent and map parts for following process. To incorporate the historical information into the current, we employ map-map and agent-scene cross-attention modules with Multi-Head Attention (MHA) blocks for map and agent features, respectively:

$$F_{\mathrm{m}}^t = \mathrm{MHA}(\mathrm{Q} = F_{\mathrm{m}}^t,\, \mathrm{K} = \tilde{F}_{\mathrm{m}}^{t-1},\, \mathrm{V} = \tilde{F}_{\mathrm{m}}^{t-1}),$$
$$F_{\mathrm{a}}^t = \mathrm{MHA}(\mathrm{Q} = F_{\mathrm{a}}^t,\, \mathrm{K} = \tilde{F}_{\mathrm{s}}^{t-1},\, \mathrm{V} = \tilde{F}_{\mathrm{s}}^{t-1}), \qquad (2)$$
$$F_{\mathrm{s}}^t = \mathrm{Concatenate}(F_{\mathrm{a}}^t,\, F_{\mathrm{m}}^t),$$

where map features only interact with each other across time to enrich map elements, while agent features aggregate the overall historical scene context for comprehensive scene understanding.

Given the sequential nature, this past context-referenced feature will be further propagated to future timestamps. Simultaneously, it serves as the input to the decoder for trajectory prediction at the current timestamp.

### 3.2.3 Agent trajectory stream

In addition to referencing scene context, we enhance trajectory forecasting by establishing temporal relationships to achieve further improvement. This involves leveraging the inherent temporal continuity and smoothing nature of trajectories. This is accomplished through the agent trajectory stream, which is equipped with a memory bank for storing historical trajectories, enabling temporal relaying.

To maintain a set of $n$ historical trajectories for each agent of interest, we design the trajectory memory bank as $\mathcal{M}(a) = \{(y_1, f_1), (y_2, f_2), ..., (y_n, f_n)\}$, where $y_i$ denotes predicted trajectories projected onto the global coordinate system, and $f_i$ represents corresponding mode features recording historical motion information. Unlike the detection or tracking tasks, where position changes of objects typically remain relatively consistent, the future motion patterns (under local system) involves significant changes due to variations in road conditions. Therefore, directly decoding with historical memory query features is inappropriate. Considering that the simple decoder can provide satisfactory predictions in practice, modifying initial predictions using the memory bank proves more effective.

Before refinement, we align the saved trajectories and mode features with the current coordinate system. While handling features consistently with the above, it is crucial to align trajectories in Euclidean space for more precise comparisons. Concerning the trajectory $y_i \in \mathbb{R}^{K \times 2}$ (with $K$ being the frame steps of the trajectories), we compute the transformed trajectory $\tilde{y}_i$ as

$$\tilde{y}_i = \mathcal{R}(\theta) \cdot (y_i - y_i^{\mathrm{ori}})^{\mathrm{T}},\ \text{where}\ y_i^{\mathrm{ori}} = y_i[\Delta t \cdot q]. \qquad (3)$$

Here, $\mathcal{R}(\theta)$ is the rotation matrix for the current heading angle, and $y_i^{\mathrm{ori}}$ denotes the trajectory origin chosen based on the time difference $\Delta t$ and sampling frequency $q$. Recognizing that memory trajectories may originate from different historical times, their origins also differ.

After transformation, trajectories whose latter part resembles the former part of current predictions contribute more to the refinement process. To aggregate memory information accordingly, we update the current mode features with a lightweight Transformer Decoder utilizing Trajectory Embedding (TE) to measure spatial similarity and replace the original positional embedding. Then, we further propagate updated modes into a MLP module to generate offsets and update initial predictions. This procedure is defined as follows:

$$F_{\mathrm{mo}} = \mathrm{MHA}(\mathrm{Q} = F_{\mathrm{mo}} + \mathrm{TE}(Y_{\mathrm{mo}}),\, \mathrm{K} = \tilde{F}_{\mathrm{b}} + \mathrm{TE}(\tilde{Y}_{\mathrm{b}}),\, \mathrm{V} = \tilde{F}_{\mathrm{b}}),$$
$$Y_{\mathrm{mo}} = \mathrm{MLP}(F_{\mathrm{mo}}) + Y_{\mathrm{mo}}, \qquad (4)$$

where $F_{\mathrm{mo}}$ and $Y_{\mathrm{mo}}$ are mode features and initial predicted trajectories in the current context, $\tilde{F}_{\mathrm{b}}$ and $\tilde{Y}_{\mathrm{b}}$ represent aligned features and trajectories retrieved from the memory bank. Besides, we employ a single layer MLP to embed the flattened trajectories as the TE.

Ultimately, we save the refined trajectories (projected onto the global system) and features while removing outdated ones (first in first out). This updated trajectory memory will be further passed down for future timestamps. Importantly, due to the generation of only a small number of motion modes, we do not apply any filtering to ensure the multimodality and diversity of the bank.

### 3.3 Model training

During the training process, we supervise the estimated trajectories using the regression loss $\mathcal{L}_{\mathrm{reg}}$ and the associated probabilities through the classification loss $\mathcal{L}_{\mathrm{cls}}$. Additionally, we introduce the

Table 1: Performance comparison on *Argoverse 2 test set* in the official leaderboard. For each metric, the best result is in **bold** and the second best result is underlined. "-": Unreported results; "†": Methods that use model ensemble trick. RealMotion-I refers to the independent variant of our model without data reorganization and stream modules, simply taking the original trajectory as input to forecast the motion like previous methods.

| Method | $minFDE_1$ | $minADE_1$ | $minFDE_6$ | $minADE_6$ | $MR_6$ | $b\text{-}minFDE_6$ |
|---|---|---|---|---|---|---|
| HDGT [19] | 5.37 | 2.08 | 1.60 | 0.84 | 0.21 | 2.24 |
| THOMAS [14] | 4.71 | 1.95 | 1.51 | 0.88 | 0.20 | 2.16 |
| GoRela [7] | 4.62 | 1.82 | 1.48 | 0.76 | 0.22 | 2.01 |
| HPTR [48] | 4.61 | 1.84 | 1.43 | 0.73 | 0.19 | 2.03 |
| QML† [34] | 4.98 | 1.84 | 1.39 | 0.69 | 0.19 | 1.95 |
| Forecast-MAE [5] | 4.36 | 1.74 | 1.39 | 0.71 | 0.17 | 2.03 |
| TENET† [42] | 4.69 | 1.84 | 1.38 | 0.70 | 0.19 | 1.90 |
| BANet† [45] | 4.61 | 1.79 | 1.36 | 0.71 | 0.19 | 1.92 |
| GANet [39] | 4.48 | 1.78 | 1.35 | 0.73 | 0.17 | 1.97 |
| SIMPL [47] | - | - | 1.43 | 0.72 | 0.19 | 2.05 |
| Gnet† [11] | 4.40 | 1.72 | 1.34 | 0.69 | 0.18 | 1.90 |
| ProphNet [41] | 4.74 | 1.80 | 1.33 | 0.68 | 0.18 | **1.88** |
| QCNet [51] | 4.30 | 1.69 | 1.29 | **0.65** | 0.16 | 1.91 |
| RealMotion-I | 4.42 | 1.73 | 1.38 | 0.70 | 0.18 | 2.01 |
| **RealMotion** | **3.93** | **1.59** | **1.24** | 0.66 | **0.15** | 1.89 |

refinement loss $\mathcal{L}_{\mathrm{refine}}$ to guide the learning of predicted trajectory offsets within our agent trajectory stream. The overall loss $\mathcal{L}$ combines these individual losses with equal weights, formulated as follows:

$$\mathcal{L} = \mathcal{L}_{\mathrm{reg}} + \mathcal{L}_{\mathrm{cls}} + \mathcal{L}_{\mathrm{refine}}, \tag{5}$$

For $\mathcal{L}_{\mathrm{reg}}$ and $\mathcal{L}_{\mathrm{refine}}$, we employ the smooth-L1 loss, while the cross-entropy loss is utilized for $\mathcal{L}_{\mathrm{cls}}$.

## 4 Experiments

### 4.1 Experimental settings

**Datasets and metrics** We assess the performance of our method using the Argoverse 1 [3] and Argoverse 2 [43] motion forecasting datasets in both single-agent and multi-agent settings. The Argoverse 1 dataset comprises 323,557 sequences from Miami and Pittsburgh, while the Argoverse 2 dataset contains 250,000 scenes spanning six cities. In the Argoverse 1 dataset, predictors are tasked with forecasting 3 seconds of future trajectories for agents based on 2 seconds of historical observations. In contrast, the Argoverse 2 dataset offers improved data diversity, higher data quality, a larger observation window of 5 seconds, and an extended prediction horizon of 6 seconds. Additionally, both datasets have a sampling frequency of 10 Hz.

We employ standard benchmark metrics, encompassing minimum Average Displacement Error ($minADE_k$), minimum Final Displacement Error ($minFDE_k$), Miss Rate ($MR_k$), and brier minimum Final Displacement Error ($b - minFDE_k$), across six prediction modes designed for the single-agent setting. Further details can be found in the appendix.

**Implementation details** We train our models using the AdamW [24] Optimizer with a batch size of 32 per GPU for 60 epochs. Our model is trained end-to-end with a learning rate of 0.001 and a weight decay of 0.01. The latent feature dimension is set to 128. Following [5, 32], we consider only agents and lane segments within a 150-meter radius of the focal agent. For the samples in Argoverse 2, we split the whole scene into 3 segments, each with a historical observation window of 3s and a same prediction horizon of 6s as the original. As for the Argoverse 1, we set the historical window of 1s. To fully utilize historical information, we compute gradients and perform back propagation for all segments. Moreover, the RealMotion-I is trained and evaluated with a common configuration without dataset reorganization and stream modules.

Table 2: Performance comparison on *Argoverse 1 validation set*.

| Method | $minADE_6$ | $minFDE_6$ | $MR_6$ |
|--------|-----------|-----------|--------|
| LaneRCNN [44] | 0.77 | 1.19 | 0.08 |
| DenseTNT [15] | 0.73 | 1.05 | 0.10 |
| mmTransformer [23] | 0.71 | 1.15 | 0.11 |
| LaneGCN [22] | 0.71 | 1.08 | - |
| PAGA [8] | 0.69 | 1.02 | - |
| DSP [46] | 0.69 | 0.98 | 0.09 |
| ADAPT [1] | 0.67 | 0.95 | 0.08 |
| HiVT [52] | 0.66 | 0.96 | 0.09 |
| R-Pred [6] | 0.66 | 0.95 | 0.09 |
| HPNet [35] | 0.64 | **0.87** | **0.07** |
| **RealMotion** | **0.61** | 0.91 | **0.07** |

Table 3: Performance comparison on *Argoverse 2 Multi-agent test set* in the official leaderboard.

| Method | $avgMinFDE_1$ | $avgMinADE_1$ | $avgMinFDE_6$ | $avgMinADE_6$ | $actorMR_6$ |
|--------|--------------|--------------|--------------|--------------|-------------|
| FJMP [31] | 4.00 | 1.52 | 1.89 | 0.81 | 0.23 |
| Forecast-MAE [5] | 3.33 | 1.30 | 1.55 | 0.69 | 0.19 |
| Gnet [11] | 3.05 | 1.23 | 1.46 | 0.69 | 0.19 |
| RealMotion | **2.87** | **1.14** | **1.32** | **0.62** | **0.18** |

## 4.2 Comparison with state of the art

We first compare the performance of our RealMotion with several top-ranked models on the Argoverse 2 [43] motion forecasting benchmark. The results on the test split are presented in Tab. 1. RealMotion has far outperformed most of previous approaches. Concretely, our method stands distinctly ahead of other methods in terms of $minFDE_1$ and $minADE_1$, showing performance enhancements of 8.60% and 5.91% relative to QCNet, respectively. We also get the almost top 2 place for other metrics. Compared to our independent variant RealMotion-I, the proposed method exhibits significant performance improvements across all metrics, which conclusively demonstrates the effectiveness of our designs. Then, we compare the performance of RealMotion on the Argoverse 1 benchmark, with the results of the validation split presented in Tab. 2. It is shown that our method also achieves a decent performance, especially for $minADE_6$. We also provide our ensemble and multi-agent results in Appendix B.

## 4.3 Multi-agent quantitative results

In the multi-agent setting, predictors are required to jointly estimate the future trajectories of all interested agents, which is crucial for the comprehensive perception of the driving scenario. Therefore, we also evaluate our RealMotion on the Argoverse 2 Multi-agent test set to prove the effectiveness, and provide simple results as shown in Tab. 3. Although not integrated with specialized designs for multi-agent forecasting like [26, 1], our model also demonstrates superior performance compared to recent works owing to our sequential techniques.

## 4.4 Ablation study

We conduct ablation studies on the Argoverse 2 validation split for the single-agent setting to examine the effectiveness of each component in RealMotion. We adopt the default experiment settings following Sec. 4.1 to perform ablation in this section.

**Effects of components.** As shown in Tab. 4, we assess the effectiveness of each component in our network. The first row (ID-1) represents the independent framework RealMotion-I, which is the same as reported in Tab. 1. First, our data reorganization approach extends dataset at no extra cost and enables the exhaustive utilization of temporally continuous motion, resulting in a noticeable improvement as shown in the second row. Then, our proposed two streams can better learn temporal

Table 4: Ablation study on the core components of RealMotion on the *Argoverse 2 validation set*. "Con. Data" indicates the processed continuous scenes. "SC Strm" and "AT Strm" indicate our proposed scene context stream and agent trajectory stream, respectively.

| ID | Con. Data | SC Strm | AT Strm | $minFDE_1$ | $minADE_1$ | $minFDE_6$ | $minADE_6$ | $MR_6$ | $b\text{-}minFDE_6$ |
|---|---|---|---|---|---|---|---|---|---|
| 1 | | | | 4.499 | 1.793 | 1.423 | 0.721 | 0.185 | 2.054 |
| 2 | ✓ | | | 4.397 | 1.722 | 1.357 | 0.687 | 0.169 | 2.001 |
| 3 | ✓ | ✓ | | 4.129 | 1.648 | 1.344 | 0.678 | 0.160 | 1.987 |
| 4 | ✓ | | ✓ | 4.194 | 1.667 | 1.331 | 0.673 | 0.164 | 1.976 |
| 5 | ✓ | ✓ | ✓ | 4.091 | 1.620 | 1.312 | 0.664 | 0.156 | 1.961 |

Table 5: Ablation study on (a) (left) the Feature Alignment and Trajectory Embedding and (b) (right) the gradient steps and split points. For (a), "C.A." and "T.A." represent the feature alignment modules used in the Context Referencing and the Trajectory Relaying blocks. "T.E." represents the Trajectory Embedding. For (b), "Grad Steps" indicates the number of steps we take to compute the gradient. "Split Pts" indicates the split points used to divide the trajectory.

| C.A. | T.A. | T.E. | $minFDE_6$ | $minADE_6$ | $MR_6$ |
|---|---|---|---|---|---|
| | | | 1.334 | 0.681 | 0.163 |
| ✓ | | | 1.328 | 0.674 | 0.160 |
| | ✓ | | 1.326 | 0.673 | 0.158 |
| ✓ | ✓ | | 1.324 | 0.670 | 0.158 |
| ✓ | ✓ | ✓ | 1.312 | 0.664 | 0.156 |

| Steps | Split Pts | $minFDE_6$ | $minADE_6$ | $MR_6$ |
|---|---|---|---|---|
| 1 | (30, 40, 50) | 1.420 | 0.716 | 0.175 |
| 2 | (30, 40, 50) | 1.341 | 0.681 | 0.162 |
| 3 | (30, 40, 50) | 1.312 | 0.664 | 0.156 |
| | (20, 35, 50) | 1.331 | 0.674 | 0.158 |
| | (40, 45, 50) | 1.365 | 0.692 | 0.163 |
| 5 | (30, 35, 40, 45, 50) | 1.323 | 0.668 | 0.158 |

relationships at the scene level, thereby both bringing additional improvements, represented as ID-3 and ID-4, respectively. Considering that these two streams are complementary to each other, therefore, RealMotion that involves all these sequential techniques achieves remarkable performance gains, as indicated in the final row. Additionally, in contrast to the consistent improvements observed with the data processing approach, it is worth noting that our two streams demonstrate more significant advantages in single-mode metrics compared to the six-mode metrics. As shown in Fig. 4, we attribute this phenomenon to the enhanced capability of our streams to modify less accurate trajectories.

**Effects of feature alignment and TE.** The misalignment of features might have an adverse effect on the performance when implementing feature interaction across scenes. Hence, we evaluate the impact of the alignment modules in both the Context Referencing and the Trajectory Relaying blocks in Tab. 5(a). As depicted in the first four rows, the removal of the alignment modules clearly results in a performance decline. By incorporating these modules, We have, to some extent, alleviated the negative impact of misalignment. However, executing it twice only yields marginal gains, which might be caused by the function overlap. Besides, we also analyze the performance of our proposed Trajectory Embedding in the 4th and 5th rows. The performance gains indicate that the Trajectory Embedding can facilitate the selection of similar historical trajectories from the memory bank, thereby easily constraining and refining current predictions. Despite the simplicity in the design of these modules, they both contribute to additional benefits for our network.

**Effects of split points and gradient steps.** In Tab. 5(b), we investigate the performance variations with respect to the number of gradient steps taken and the split points along historical steps. From the 1st to the 3rd row, as we progressively increase the number of steps used for gradient computation, there is a noticeable improvement in performance. This enhancement can be attributed to the fact that computing the gradient for specific steps allows us to better capture the trajectory distribution for model training. From the 3rd to the 5th row, we change the interval of split points from 5 frames to 15 frames. Accordingly, the length of historical trajectory also changes, ranging from 40 frames to 20 frames. As observed, similar trajectories occur in the sequence when using a short interval, which can significantly hinder the model to learn distinct motion patterns. Conversely, using a longer interval results in fewer historical trajectory frames available in each segment, which contradicts the optimization of one-shot forecasting. It is evident that our choices for the gradient steps and the split points are well-suited for the Argoverse 2 benchmark. Moreover, we attempt to divide the trajectory into 5 segments in the final row. An excessively long trajectory sequence imposes a

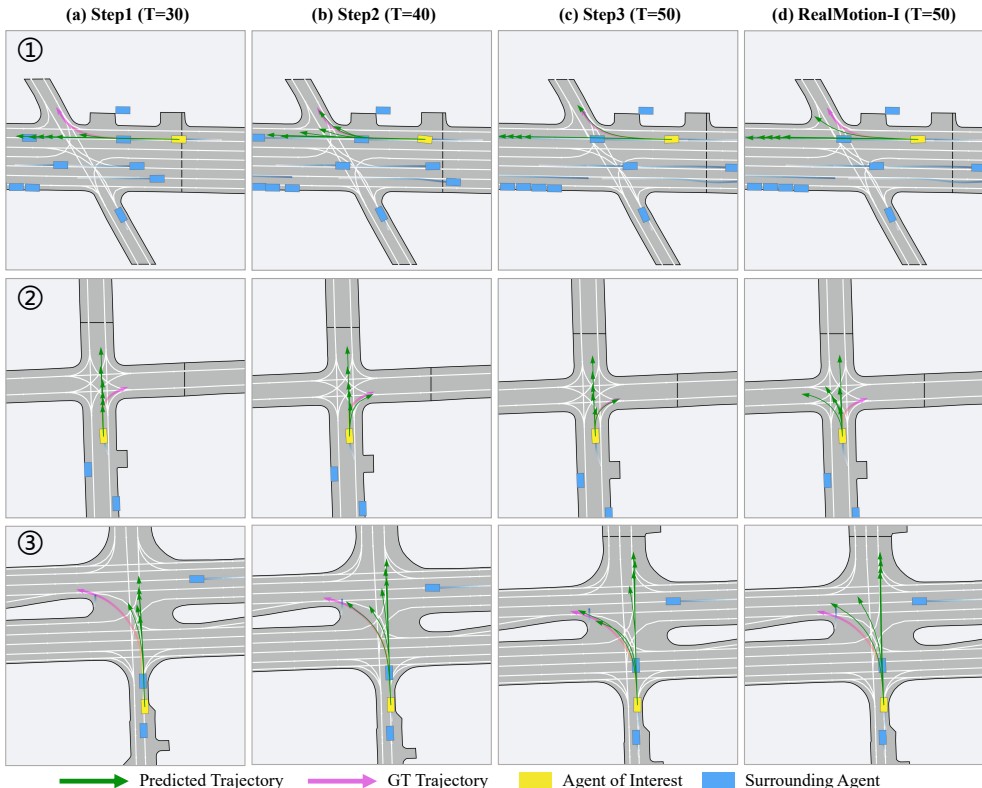

| (a) Step1 (T=30) | (b) Step2 (T=40) | (c) Step3 (T=50) | (d) RealMotion-I (T=50) |

→ Predicted Trajectory    → GT Trajectory    ▮ Agent of Interest    ▮ Surrounding Agent

Figure 4: Qualitative results on the *Argoverse 2 validation set*. The panel (a)-(c) demonstrate the progressive forecasting results of our RealMotion, where the panel (c) is the final predictions for evaluation. The panel (d) shows the one-shot forecasting of RealMotion-I.

Table 6: Comparison of model performance, inference speed, and memory size. "Latency": Inference speed. "Params": The number of parameters.

| Method | Latency | Params | $minFDE_6$ | $minFDE_1$ |
|---|---|---|---|---|
| HPTR (online)[48] | **13ms** | 15.1M | 1.43 | 4.61 |
| HPTR (offline) | 28ms | | | |
| QCNet[51] | 94ms | 7.7M | 1.29 | 4.30 |
| RealMotion-I | 16ms | **2.0M** | 1.38 | 4.42 |
| RealMotion (online) | 20ms | 2.9M | **1.24** | **3.93** |
| RealMotion (offline) | 62ms | | | |

learning burden on our framework, preventing it from focusing on temporal relationships and yielding limited improvements.

**Effects of the depth of cross-attention blocks.** As shown in Tab. 7, we explore the influence of depth variations of cross-attention unit in the Context Referencing and the Trajectory Relaying blocks. Primarily our temporal blocks are lightweight regardless of depth to ensure the efficiency and universality. a relatively deep cross-attention unit in the Context Referencing and the Trajectory Relaying blocks is necessary for processing history information and current information. We use a depth of 2 as our default setting considering its better efficiency-performance balance.

## 4.5 Efficiency analysis

Balancing performance, inference speed, and model size is important for the model deployment. We compare our RealMotion with recent representative works, which include the real-time forecasting approach HPTR [48] and the state-of-the-art approach QCNet [51]. We measure these approaches

Table 7: Ablation on the cross-attention block depth. "Params": The number of parameters.

| depth | Params | $minFDE_6$ | $minADE_6$ | $MR_6$ |
|-------|--------|------------|------------|--------|
| 1 | 2.5M | 1.348 | 0.679 | 0.165 |
| 2 | 2.9M | 1.312 | 0.664 | 0.156 |
| 3 | 3.3M | 1.328 | 0.677 | 0.159 |

and RealMotion on the Argoverse 2 test set using an NVIDIA GeForce RTX 3090 GPU, maintaining a batch size of 1 and following an end-to-end manner. As shown in Tab. 6, RealMotion has the competitive inference time and a competitive model size while achieving the best performance. It is worth noting that "online" indicates the latency for practical autonomous driving systems, which is optimized compared to the "offline" version by some efficient designs (e.g. the caching technique in [48]). As for RealMotion, we must consider three times latency for "offline" dataset, where only the final prediction is utilized for evaluation. In contrast, the forecasting results of each iteration is meaningful in "online" application.

### 4.6 Qualitative results

In Fig. 4, we present qualitative results of our network compared to the independent version on the Argoverse 2 validation set. Panels (a), (b), and (c) illustrate the forecasting results at 3s, 4s and 5s, respectively. Panel (c) displays the final results used for evaluation, which are more accurate and closer to the ground truth. By comparing panel (c) and (d), it can be seen that RealMotion remarkably outperforms the independent version. As demonstrated in the panels from (a) to (c), our RealMotion can progressively refine the estimated trajectories from coarse to fine as the motion progresses and accurately capture the possible motion intention. However, the one-shot forecasting of RealMotion-I shown in the panel (d) leads to significant estimation errors.

## 5 Conclusion

In this work, we anticipate to address the motion forecasting task from a more practical continuous driving perspective. This in essence places the motion forecasting function in a wider scene context compared to the previous setting. We further present RealMotion, a generic framework designed particularly for supporting the successive forecasting actions over space and time. The critical components of our framework are the scene context stream and the agent trajectory stream, both of which function in a sequential manner and progressively capture the temporal relationships. Our extensive experiments under several settings comprehensively demonstrate that RealMotion surpasses the current state-of-the-art performance, thereby offering a promising direction for safe and reliable motion forecasting in the rapidly evolving field of autonomous driving.

**Limitations.** A clear constraint of our data processing approach is its requirement for a sufficient number of historical frames for serialization. Consequently, it is not applicable to short-term benchmarks such as the Waymo Open Dataset, which provides only 10 frames of historical trajectory. Moreover, existing datasets typically provide limited historical information distinct from real-world settings, which inhibits our sequential designs from fully leveraging their advantages. Hence, we anticipate to integrate our framework into a sequential autonomous driving system to maximize the benefits of streaming designs in our future work.

## Acknowledgments

This work was supported in part by National Natural Science Foundation of China (Grant No. 62106050 and 62376060), Natural Science Foundation of Shanghai (Grant No. 22ZR1407500).

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

# Appendix

## A  More experimental settings

### A.1  Evaluation metrics

For single-agent evaluation, we employ standard metrics for evaluation, including minimum Average Displacement Error ($minADE_k$), minimum Final Displacement Error ($minFDE_k$), Miss Rate ($MR_k$), and brier minimum Final Displacement Error ($b - minFDE_k$). $minADE_K$ calculates the $L_2$ distance between the ground-truth trajectory of the best $K$ predicted trajectories, averaged across all future time steps. While $minFDE_k$ measures the difference between the predicted endpoints and the ground truth. $MR_k$ is the ratio of scenes where $minFDE_k$ exceeds 2 meters. To further assess uncertainty estimation performance, the metric $b - minFDE_k$ adds $(1 - \pi)^2$ to the final-step error, where $\pi$ denotes the best-predicted trajectory's probability score that the model assigns. As a common practice, $K$ is selected as 1 and 6.

For multi-agent evaluation, We use standard metrics including Average Minimum Final Displacement Error ($avgMinFDE$), Average Minimum Average Displacement Error ($avgMinADE$) and Actor Miss Rate ($actorMR$). $avgMinFDE$ is the average of the lowest FDEs for all scored actors within a scene, reflecting the prediction accuracy of a scene outcome. $avgMinADE$ represents the average of the lowest ADEs for all scored actors within a scene, indicating the general accuracy of the predicted movements. Across the evaluation set, the $actorMR$ is the proportion of missed actor (same as above).

## B  More experiments

### B.1  Model ensemble

Model ensemble, a crucial technique for enhancing the accuracy of final predictions, is employed in our approach. We utilize six sub-models trained with various random seeds and split points. Consequently, we generate 36 predicted future trajectories for each agent, and then apply k-means clustering to process them with 6 cluster centers. For each cluster group, we calculate the average of all trajectories within the group to produce the final trajectories. The results with and without model ensemble trick on the Argoverse 2 test set are shown in Tab. 8.

Table 8: Performance comparison between results with and without ensemble on *Argoverse 2 test set* in the official leaderboard. For each metric, the better result is in **bold**.

| RealMotion | $minFDE_1$ | $minADE_1$ | $minFDE_6$ | $minADE_6$ | $MR_6$ | $b$-$minFDE_6$ |
|---|---|---|---|---|---|---|
| w/o ensemble | 3.93 | 1.59 | 1.24 | 0.66 | 0.15 | 1.89 |
| w/ ensemble | **3.87** | **1.55** | **1.18** | **0.63** | **0.13** | **1.78** |

### B.2  Generality evaluation with integrating RealMotion

For generality test, we integrate our RealMotion data with the state-of-the-art method QCNet [51]. Due to the big size of QCNet, the scene context and agent trajectory streams have to be excluded for memory constraint. With minimal alterations applied to this prior model, we observe a noticeable improvement. The results are shown in Tab. 9.

Table 9: Generality evaluation with integrating RealMotion.

| Method | $minFDE_6$ | $minADE_6$ | $MR_6$ |
|---|---|---|---|
| QCNet [51] | 1.27 | 0.73 | 0.16 |
| QCNet w/ RealMotion | **1.24** | **0.71** | **0.15** |

# C   More qualitative results

We provide more qualitative results of our framework in Fig. 6 and Fig. 7 on the Argoverse 2 validation set.

# D   Failure cases

Although our RealMotion has achieved outstanding performance on motion forecasting benchmarks, it still has some failure cases. We analyze these typical cases and provide qualitative results to help readers understand the circumstances under which our model may fail. This analysis will aid future work in developing a more powerful and robust algorithm, as shown in Fig. 5.

## D.1   Case 1: Complex map topology

In the first row of Fig. 5, the agent requires to navigate through a complex intersection to one of the roads, but the model fails to predict this possible driving behavior and just anticipates the agent to drive straight ahead. This may be caused by the lack of a comprehensive understanding of the complex map topology and the unbalanced distribution of driving data. In most scenarios, agents tend to exhibit only trivial behaviors, such as moving straight ahead at a nearly constant velocity. Consequently, this raises issues regarding data balance, and the figures indicate that our model is more likely to make mistakes when it comes to turning.

## D.2   Case 2: Subjective driving behavior

In the second row of Fig. 5, the vehicle is expected to park on the side of the road, which is a kind of subjective driving behaviors. However, the model only predicts that the vehicle will keep going ahead. To improve the forecasting of such cases, we could enhance the interaction between the model with additional intentions of human and incorporate more information, such as visual cues like turn signals and parking spaces.

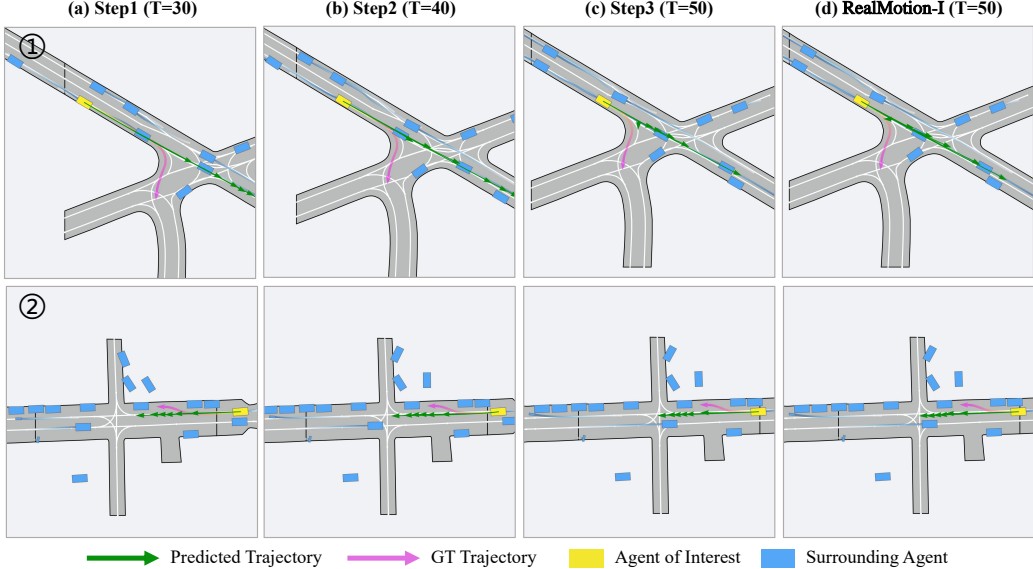

Figure 5: Failure cases. In the first row, the model fails to predict the turning behavior at complex intersections, while in the second row, it fails to predict the parking behavior.

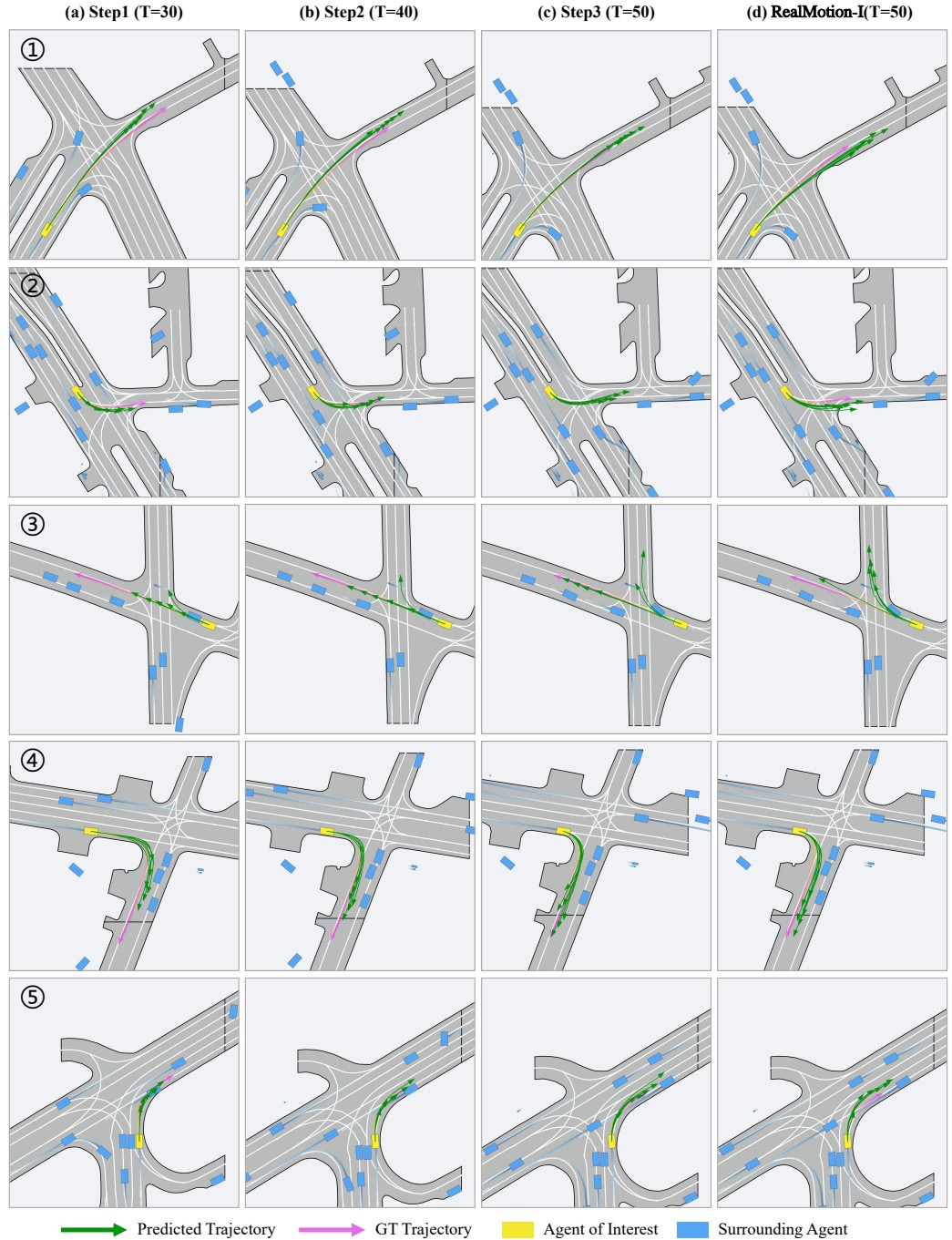

Figure 6: More qualitative results.

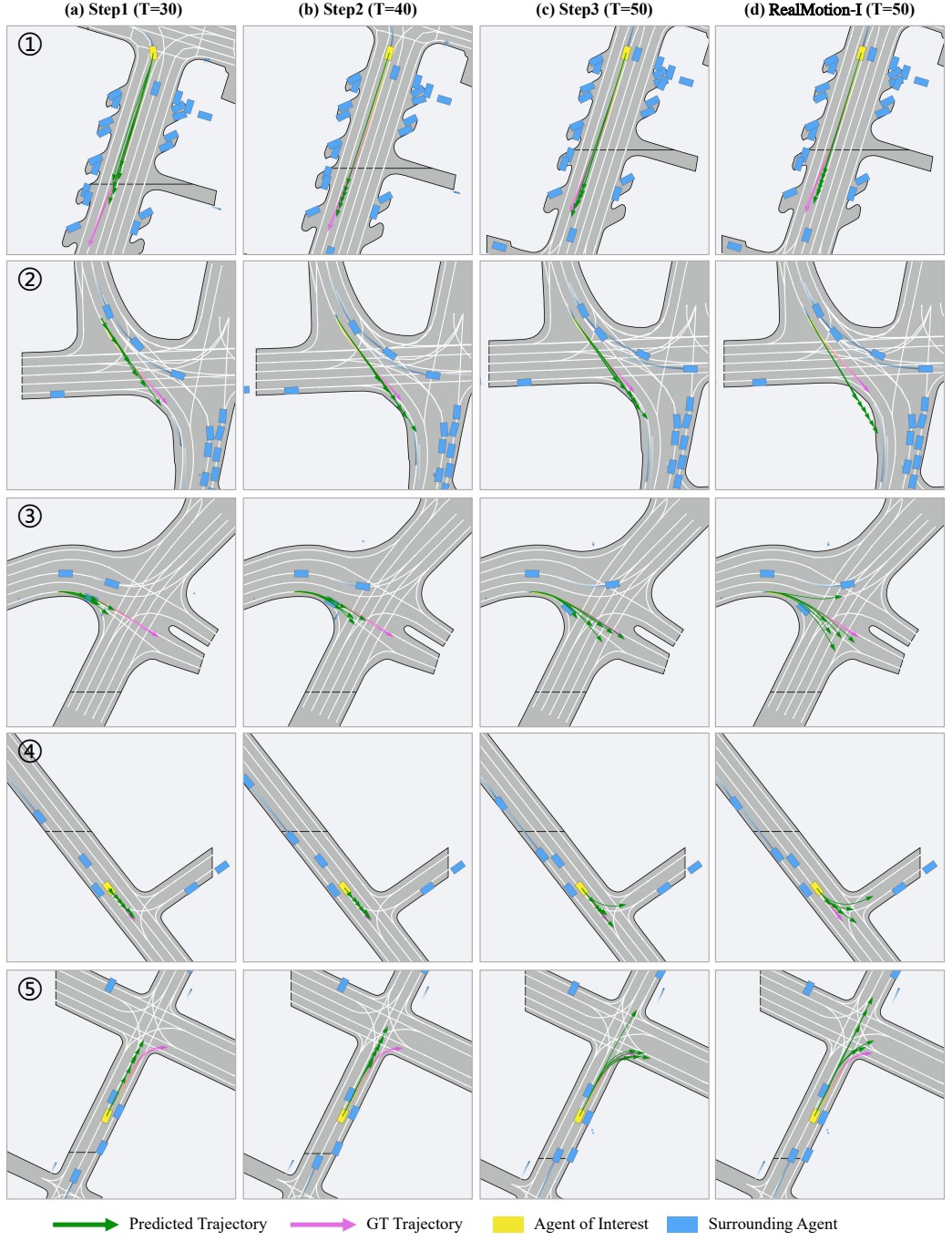

Figure 7: More qualitative results.

