# OpenReview forum: "Motion Forecasting in Continuous Driving"
_NeurIPS.cc/2024/Conference — NeurIPS 2024 spotlight_

### Official Review · Reviewer_u4R4 · 2024-07-11

**Soundness:** 4
**Presentation:** 2
**Contribution:** 4
**Rating:** 7
**Confidence:** 4

**Summary:**

This paper proposes a model for motion forecasting that models the continuous stream of world state over sequential timesteps.  This is in contrast to most/all other works which do an indepedent prediction of the future for each timestep based on a fixed window of history, without context of previous model belief states or outputs from previous time steps.  One stream of data is scene context, represented as embeddings, where context is fused with current accounting for transformation of the scene due to ego motion.  The second stream is agent trajectories, where a history of n previous predicted trajectories for each agent is maintained explicitly in a memory bank, and fused into the current time steps prediction as additional signal.

They do thorough ablations on Argoverse benchmark dataset, showing compelling evidence that the streaming architecture is beneficial.  Furthermore, they exceed SOTA on most metrics in the benchmark. When I checked today (mid July), they are ranked 7th overall in the leaderboard - https://eval.ai/web/challenges/challenge-page/1719/leaderboard/4098

**Strengths:**

This paper addresses an important problem often ignored in these sanitized datasets: that these models in real world  application in an AV system run in a sequential streaming fashion, and (to my knowledge) all other works operate on each time frame independently (given a fixed window of historical state).

The proposed solution itself is a good one, transforming old belief state into current reference frames in both embedding space and with spatial transformations where appropriate.  At the same time, the complexity of the architecture is kept reasonably low, with each contribution justified.

Results are very strong and well justified with thorough ablations.

**Weaknesses:**

Minor, but I had some difficulty understanding some of the explanations:

 - Data reorganization / Figure 2: "we select several split points Ti along historical frame steps" --> how are they selected? what are "historical frame steps"? Can you more simply say this as, for example, "We split sequences evenly into sub-sequences"?

 - Around L158: you started using the term "modes" for the first time, where I think you should just say trajectories, or be more careful defining the tensors F_{mo} and Y_{mo} making clear their dimensionality.

 - You never clearly state the output representation of your model (which I assumed to be a weighted trajectory set). Without being more clear here, Section 3.3. Model training is sort of meaningless - for example, what is the classification loss classifying? (I'm 99% sure it's a softmax distribution over trajectory modes, but this is not said anywhere.)

**Questions:**

See weaknesses

---

> ### Author Rebuttal · Authors · 2024-08-06
>
> We thank the reviewer for the detailed review as well as the suggestions for improvement. Our response to the reviewer’s comments is below:
>
> Apologies for these unclear explanations.
>
> + **Data reorganization:**
> As shown in Fig. 2(b), the dashed part refers to historical time steps (50 historical steps in AV2), and we select several split points on this part. We consider an original trajectory (the gray arrow) as a sequence, and our processing approach generates several shorter trajectories (sub-sequences). Besides, the split points are manually specified in our method and discussed in ablation Tab.4(b).
>
> + **Mode:**
> All correct! Multiple trajectories are predicted for each agent to enable comprehensive motion possibilities, in which case $mode$ is commonly used to represent each prediction. We will make it clear. Besides, the dimension of $Y_{mo}$ is $[B, N_a, N_{mode}, 2]$.
>
> + **Model output:**
> The output is several predicted trajectories with corresponding probabilities, as the reviewer understood. While providing the types of loss functions in Line 177, we indeed overlooked the concrete input items. We will clarify this.

---

### Official Review · Reviewer_GAo1 · 2024-07-11

**Soundness:** 2
**Presentation:** 2
**Contribution:** 2
**Rating:** 6
**Confidence:** 4

**Summary:**

This submission tackles the task of trajectory forecasting in autonomous driving.
In particular, it proposes improvements in two aspects:

1/ Data reorganization: Current datasets are artificially split into non-overlapping segments. The authors propose to reorganize the data to have overlapping windows.

2/ Relaying context and predictions: since the sub-scenes now overlap, they can use context and predictions from the previous timesteps (corresponding to a different sub-scene) to improve predictions at the current timestep.
Relaying context serves to provide more information about the surroundings (an agent could have been missed in the current frame for instance), and relaying the predictions would smooth out the predictions across time.

They demonstrate the validity of their approach on Argoverse benchmarks, achieving state-of-the-art performances.

**Strengths:**

- The proposed improvements are interesting, sound, and motivated.
- The data reorganization, I believe, is a simple idea that would that should be pushed in the motion forecasting community to build a new standard that better leverage the data. It can also immediately be applied to other models and hopefully improve performances.
- The relaying through time helps achieve strong results either with their architecture or on top of QCNet.

**Weaknesses:**

- The results are not very conclusive:
  - Compared to  QCNet, except for $ADE_1$/$FDE_1$ (which are not the main metrics usually considered for motion forecasting) the improvements are small. It would be useful to give more insight into why only these metrics improve. We also don't have the $ADE_1$/$FDE_1$ values of QCNet + RealMotion so we don't know if these findings are consistent across models.
  - The comparisons are not fair in terms of available information. Because of the propagation, RealMotion has access to more past information both in the past trajectories and agents, and the past map. A naïve way to integrate more past trajectories would be to just use longer past history. This should be a minimal baseline or ablation in the paper. If we also wanted to add agents seen in the past but not at the current timestep, it could be done with some masking or extrapolation scheme for missing frames.
Remains the past map information, for which it's indeed more complicated to have a naïve baseline for a fair comparison, as simple aggregation could drastically raise the computation footprint so I can understand why the authors wouldn't want to test this baseline.
- The loss functions, and in particular $\mathcal{L}_{refine}$ are not detailed enough. I can guess where they apply but it would be good if that was stated explicitly.
- Figure 2 is unclear to me. Perhaps it could need more captions or a rework.
- I think it would be easier to interpret the results if the architecture of RealMotion would be more clearly delivered, perhaps in a Figure  (in Appendix?) that represent the full architecture detailing in particular how the different elements interact in the encoder and the decoder that are not given in Figure 3.

**Questions:**

- Why do we see drastic improvement on ADE/FDE_1 but the other improvements are very tame? Does that imply most of the improvements are to the scoring function so that RealMotion gets a better top 1? If that's the case, what could be the mechanisms that explain this behavior?

Otherwise, I think this paper rightfully questions some practices of motion forecasting in autonomous driving. Generalizing something along the lines of the proposed data reorganization would be very helpful for the community. However, I think the experiments on relaying might not be fair enough and need to be compared to simple baselines with more/better past history. We would also need a few more numbers to see if most of the findings and recommendations have a good chance at being consistent accross models.

**Limitations:**

- Limitations are discussed in the Appendix.

- Broader societal impacts do not seem explicitly addressed despite what has been reported in the checklist. I'm not sure there are obvious uncontroversially negative societal impacts to be discussed though.

---

> ### Author Rebuttal · Authors · 2024-08-06
>
> We thank the reviewer for the detailed review as well as the suggestions for improvement. Our response to the reviewer’s comments is below:
>
> **Q1: Drastic improvement on $ADE_1/FDE_1$.**
>
> By comparing QCNet and other methods (such as ProphNet), it can be seen this phenomenon is common. This is due to different calculation of metrics: the metrics@1 is the error for the one selected trajectory according to the score, while the metrics@6 is the minimum error among 6 predicted trajectories. Hence, there is more room for improvement for the metrics@1.
>
> Regarding the comparison between QCNet and ours, the improvement in $FDE_{6}$ is still salient. Besides, we provide the complete comparisons below. Since we can only integrate our restructured data, please refer to Tab.3 row 1 and 2 for comparison.
>
> |Method|$minFDE_1$|$minADE_1$|$minFDE_6$|$minADE_6$|
> |:---:|:---:|:---:|:---:|:---:|
> |QCNet|4.34 | 1.69 | 1.27 | 0.73|
> |QCNet w/ data|4.26 | 1.64 | 1.24|0.71|
>
>
> **Q2: The comparisons are not fair.**
>
> We do NOT use more past information compared to existing methods, but just process past information in a different way with the same datasets. Taking existing methods on Argoverse 2 as an example, they directly process each independent scenes with 50 historical frames, while we split each scene into several continuous small scenes with less past information (30 historical frames), then processing them in a streaming fashion. Hence, existing methods and RealMotion all use the same past information, making a fair comparison.
>
> **Q3: Loss functions.**
>
> Apologies for unclear expression. We will add the formulation for these losses. Especially, $L_{\rm refine}$ is utilized to supervise the refined trajectories $Y_{\rm mo}$ mentioned in Eq(4), which can be formulated as
> $L_{\rm refine} = {\rm smoothL1}(Y_{\rm mo}, Y_{\rm gt})$, where $Y_{\rm gt}$ refers to the ground truth.
>
> **Q4: Fig.2.**
>
> Fig.2 depicts how we process existing datasets with independent scenes into a continuous format to simulate real-world situations. We will remove some redundant elements to make it clearer.
>
> **Q5: The architecture of RealMotion**
>
> To ensure the generalizability of our proposed modules, as most current methods we adopt a standard forecasting framework, which consists of an agent encoder and a map encoder to respectively encode agent and map features, a Transformer encoder to implement interaction, and a decoder to generate trajectories and corresponding probabilities. The figure is shown in PDF file Fig.1.

---

> ### Comment · Reviewer_GAo1 · 2024-08-07
>
> Thank you for your answers and your clarifications. After going through the paper in light of the provided answers, it seems I indeed misunderstood an important part of the submission.
>
> To make sure I'm getting it right this time, would you agree your working hypothesis could be sumarized as follows?
> - Given a large history (of 50 frames), it is better to have a model that operate on a smaller sequence (30 frames) + an embedding of the past that you can obtain using the same model on earlier frames, than directly on the full (50 frames) history like QCNet does.

---

> ### Author Response · Authors · 2024-08-08
>
> Thanks for the quick response.
>
> Exactly. This is more than just simply considering longer historical frames like previous methods, otherwise, our method would not exceed previous alternatives. Our formulation allows to capitalize the sequential data more effectively in real world scenarios, not only emphasizing a longer historical scene context but also leveraging historical predictions, which cannot be achieved by simply increasing historical frames. We will further clarify in the revised version.

---

> ### Author Response · Authors · 2024-08-11
>
> Dear Reviewer GAo1
>
> Thanks again for the valuable comments. We believe our responses addressed all the questions/concerns. It would be great if the reviewer can kindly check our responses and provide feedback with further questions/concerns (if any). We would be more than happy to address them. Thank you!
>
> Best wishes,
>
> Authors

---

> ### Comment · Reviewer_GAo1 · 2024-08-12
>
> Indeed, after going again through the submission with the new information from the rebuttal and the reviews, I find my concerns has been adressed properly in the rebuttal.
> Moreover, I believe the reasons for my initial wrong impression have also been mostly adressed thanks to the discussions between the authors and Reviewer fgWb. Since the authors have commited to make the necessary change, I'm happy to increase my rating.

---

> > ### Author Response · Authors · 2024-08-12
> >
> > We appreciate the reviewer's time for reviewing and thanks for the recognition.

---

### Official Review · Reviewer_fqqC · 2024-07-11

**Soundness:** 3
**Presentation:** 3
**Contribution:** 3
**Rating:** 6
**Confidence:** 4

**Summary:**

This paper proposes a framework named "RealMotion" highlighting the importance of continuous motion forecasting in autonomous driving. From the formulation perspective, RealMotion investigates predicting trajectories in a continuous sequence of timestamps instead of previous independent predictions. From the methodology perspective, RealMotion proposes the "relaying" modules that propagate historical scene and agent contexts into the current frame to enhance prediction. Finally, RealMotion achieves top performance on Argoverse benchmarks and proves the importance of continuous contexts.

**Strengths:**

1. This paper investigates a foundational challenge in autonomous driving that is overlooked by previous people -- how to model the continuous contexts in motion forecasting. This problem is reasonable and is also something I would love to follow and work on.

2. The methods proposed by RealMotion are intuitive and reasonable. Relaying the historical context to current frames to enhance motion forecasting is indeed a novel contribution to autonomous driving.

3. The experiments prove the effectiveness of continuous contexts. In addition to the state-of-the-art performance, the comparison between RealMotion-I and RealMotion in Table 1 makes it a strong argument.

**Weaknesses:**

1. The authors have missed some closely related works, such as streaming motion forecasting (e.g., Pang et al.) and other end-to-end autonomous driving (e.g., Gu et al.), where motion forecasting is continuous. For instance, Pang et al. also discover the setbacks of predicting trajectories on independent timestamps, formulate the "streaming" forecasting task, and propose a differentiable filter to refine the predicted trajectories. I acknowledge the newly designed modules by RealMotion, but I would expect the authors to have a better discussion of these relevant efforts.

Pang et al. Streaming Motion Forecasting for Autonomous Driving. IROS 2023.
Gu et al. ViP3D: End-to-end Visual Trajectory Prediction via 3D Agent Queries. CVPR 2023.

2. RealMotion is also closely related to QCNet and based on its code. QCNet already has the interface of continuous motion forecasting and leverages a GRU to deal with the temporal contexts in queries. First of all, I would like the authors to clarify or discuss in the paper about the difference/improvement to QCNet more explicitly. In addition, I am also curious if it is possible to support that the "relaying" modules in RealMotion is better than QCNet's end-to-end mechanism. For instance, is it possible to plug the relaying module to QCNet and make a comparison?

3. I would like the authors to develop some metrics to analyze the improvement in the trajectories. More specifically, we wish to analyze "where does RealMotion improve the trajectory quality?" Since the main argument of this paper is "continuity," I am curious if the author can use a "fluctuation" measure like Pang et al. to prove that the trajectory smoothness is indeed improved. It is also encouraged to show some visual comparisons if space is allowed.

**Questions:**

See weakness above.

**Limitations:**

The authors have discussed limitations. No additional limitations from my side.

---

> ### Author Rebuttal · Authors · 2024-08-06
>
> We thank the reviewer for the detailed review as well as the suggestions for improvement. Our response to the reviewer’s comments is below:
>
> **Q1: Missing related work.**
>
> Thanks. We will add them.
>
> **Q2: Comparison with QCNet.**
>
> To ensure efficiency, our method adopts an agent-centric design different from QCNet. As mentioned with QCNet only the query-centric design is good at continuous motion forecasting and GRU is employed for fusing continuous predictions in the current scene. Hence, it should still be considered an independent method.
>
> We have tried to integrate our stream modules with QCNet, but unfortunately, it cannot be trained in our environment due to huge memory overhead. We hence just provided the results using our reorganized data in Appendix B.3.
>
> **Q3: Continuity metrics.**
>
> Following Pang et al., we evaluate the $fluctuation$ for RealMotion and baseline using our reorganized data as shown below. It can be observed that our method certainly improves the $fluctuation$ metric, with the Trajectory Stream playing a particularly important role.
>
> ||baseline|baseline+scene stream|baseline+traj stream|RealMotion|
> |:---:|:---:|:---:|:---:|:---:|
> |$fluctuation$|0.385|0.371|0.354|0.347|
>
> We also provide some qualitative results in PDF file Fig. 2, which demonstrates that RealMotion can better maintain temporal consistency of trajectories.

---

> > ### Comment · Reviewer_fqqC · 2024-08-08
> >
> > I have checked the rebuttal and the reviews from other reviewers. The added analytical experiment makes sense to me, and the added results of improving QCNet with RealMotion are also reasonable.
> >
> > I think the added visualization is good. For the revisions, I suggest the authors add some highlights pointing to the frames/branches where RealMotion has better continuity than the baseline.
> >
> > I maintain my original rating of weak accept for now.

---

> ### Author Response · Authors · 2024-08-08
>
> Thanks. We will further highlight the crucial parts.

---

### Official Review · Reviewer_fgWb · 2024-07-13

**Soundness:** 3
**Presentation:** 2
**Contribution:** 4
**Rating:** 6
**Confidence:** 4

**Summary:**

The paper introduces an approach to iteratively process temporal scene context for the purposes of agent motion prediction. This differs from traditional approaches that ingest the whole context directly.

The scene is processed in temporal chunks, where in each chunk contains agent-centric context consisting of other agents and map within 150m. Information is aggregated across chunks (i.e. over time) using two novel components:
- "Scene context stream", where agent/map features from the past frame are transformed using Motion-aware Layer Normalization [40] and cross-attended by current frame features.
- "Agent context stream", where agent predictions from the current frame are modified by attending predictions from past frames (stored in a memory bank and properly aligned).

This 'streaming-type' method obtains very strong results on the Argoverse 1 and 2 datasets, and is shown to improve two different baseline  model architectures based on [5] and [51] that consume the whole historical context in a single step. Some helpful ablations and latency studies of the proposed components is provided.

**Strengths:**

- Strong idea to stream scene context into the model, shown to be effective even with short 2-3 'chunks' of context over 2-3 seconds.
- Intuitive and well designed scene and agent context stream components that aggregate information across frames (or 'chunks'). The components are designed to model local information in agent-centric coordinates and aggregate it in a geometry-aware manner.
- Strong overall results on Argoverse 1 and 2, and when comparing the baseline models the new technique is extending.
- Mostly comprehensive related work section.

**Weaknesses:**

- In the abstract/intro, I find some of the framing to be a bit misleading. Examples:
  - "7:This significantly simplifies the forecasting task, making the solutions unrealistic to use in practice". I do not find this claim to be well supported. The current methods still ingest a significant amount of relevant context. To me the contribution of the paper is to follow a streaming paradigm and design more domain-appropriate fusion mechanisms across time than the vanilla MLP or transformer ops that are used traditionally.
  - "31 - 36: existing methods all tackle an artificially narrowed and unrealistic motion forecasting task".... The methods still ingest 2-3 seconds of context, but perhaps not as effectively as the proposed work. That does not confirm this overly strong claim however. Also the paper only demonstrates good performance with up to 3s of context, that does not prove yet that much longer context is helpful or practical.
  - To a similar point, Fig 1 is also a bit misleading: current methods would directly ingest the available scene history in the current datasets, and fuse the information, even if the information fusion is not done with the same geometric and streaming intuitions.

- In the method description, some details are unclear or could be explained better:
  - 83: "generate trajectory segments of identical length" Length can be misconstrued to be the length of the trajectories in meters, not that they are the same across chunks. This whole paragraph could be rewritten for clarity and a more helpful diagram could be shown -- illustrating what exactly is done for Argoverse 2 (one now needs to read all the way to Sec 4 to understand how this works).
  - "114: Additionally, we forecast a singular trajectory for auxiliary training purposes, focusing on the movement patterns of other agents. "
Unclear what this means exactly. Do you produce only a single trajectory prediction in past steps?
  - "“Equal length into past and future” → confusing, assume fixed time interval in both directions.
  - The L_refine loss in Eq 5 is not explained, unclear what is being refined. My guess is that this is predictions in earlier chunks
  - What is exactly RealMotion without the context stream components? This is present both in the Ablation and in Appendix B.3. How do you fuse information across time then?

- In the experimental results, some SOTA methods from the leaderboard seem to omit SEPT (ICLR 2023), which has higher numbers than RealMotion, even though it is cited. Why did you you omit it?

- A couple more ablations or experiments would be helpful.
  - It seems that the method is similar to a filter, where the current chunk only attends to previous chunk (but not to the ones before). Is my reading of the paper correct? If so, an experiment where the method can attend longer context memory with multiple frames would be interesting.
  - It would also be interesting to try pretraining this method on more data examples (those can be produced by predicting shorter futures, then you can use more of the segments). And then fine-tune on 6s futures.

**Questions:**

Main question is, do you agree with my reading that "the contribution of the paper is to follow a streaming paradigm and design more domain-appropriate fusion mechanisms across time than the vanilla MLP or transformer ops that are used traditionally. " As opposed to the strong claims in the current abstract / intro.

Why did you omit SEPT from the leaderboard results (Table 1)?

Please see my clarification or ablation/more experiment comments in the Weaknesses. I am particularly interested to understand why you did not try to attend to longer temporal context in the last chunk (if I read the paper correctly).

**Limitations:**

The same RealMotion methods tried here could be tried on the Waymo Open Dataset which also has 9 second segments, but one could take the first 3 seconds as history and predict 6 seconds, as opposed to the standard 1s / 8s split that is mentioned in the limitations.

---

> ### Author Rebuttal · Authors · 2024-08-06
>
> We thank the reviewer for the detailed review as well as the suggestions for improvement. Our response to the reviewer’s comments is below:
>
> **Q1: Some of the framing to be a bit misleading.**
>
> Apologies for clarity issue in abstract and introduction. We intent to stress the significance of streaming design in real-world settings. We will improve the clarity.
>
> As for Fig. 1, we discuss in real-world setting where scenes should be continuous, going beyond the datasets. In this case, current methods can only process independent scenes with limited history as set up in current datasets, while we advocate to utilize more abundant historical information.
>
> **Q2: Method description.**
>
> Apologies for unclear expression.
>
> + **83:** Indeed, the $length$ refers to time steps instead of meters. We will clarify this part.
>
> + **114:** It is not for past steps but for other agents. In addition to predicting the trajectories of agents of interest for evaluation, we also predict a single trajectory of other surrounding agents (not involved in evaluation) for auxiliary supervision. That is typical and beneficial for learning better motion pattern.
>
> + **Equal length:** We assume this refers to Line 83 -- that means the newly generated sub-trajectories have equal total steps, but the number of steps in the history and future can differ.
>
> + **refine loss:** $L_{\rm refine}$ is utilized to supervise the refined trajectories $Y_{\rm mo}$ mentioned in Eq(4), which can be formulated as $L_{\rm refine} = {\rm smoothL1}(Y_{\rm mo}, Y_{\rm gt})$, where $Y_{\rm gt}$ refers to the ground truth trajectories.
>
> + **RealMotion:** In ablation, RealMotion without stream modules can be considered as a one-shot forecasting models similar to current methods. This variant cannot fuse temporal information, but utilizing our reorganized data as a data augmentation method is still possible. In Appendix B.3, it should refer to only using our reorganized data, which we will clarify.
>
> **Q3: Comparison with SEPT.**
>
> We consider that SEPT is a pretrained self-supervised method orthogonal to ours. For example, it utilizes all sets (including test set) to address the disparity between train set and test set, leading to more data used than current methods and making the direct comparison unfair. A property way of using SEPT as model pretraining is likely to improve general methods like ours, and we could include them for further improvements.
>
> **Q4: More ablations**
>
> + Actually, the current chunk has attended to several preceding chunks. For the trajectory stream, we explicitly maintain a memory bank for more history predictions from a longer history. For the scene stream, since previous chunk involves earlier context, the current can also utilize a longer context to some extent through implicit propagation. Besides, We have tried to propagate two frames of scene context as shown below:
>
>     ||Latency|Memory|$minFDE_6$|$minADE_6$|
>     |:---:|:---:|:---:|:---:|:---:|
>     |2 frames|23ms|1.8G|1.28|0.64|
>     |1 frames|20ms|1.4G|1.31|0.66|
>
>     This strategy can indeed bring slight improvements, but we finally did not adopt more frames due to considerations of simplicity, efficiency and memory overhead, especially in complex scenarios such as intersections.
>
> + It is an excellent insight! To make the data more aligned with the forecasting task, we did not process future trajectories (such as Waymo mentioned below) in this paper, but it is indeed worth integrating and exploring distinct motion tasks (simulation, prediction and planning) and datasets. We intend to further extend our model in future work.

---

> ### Comment · Reviewer_fgWb · 2024-08-10
>
> Authors have answered my technical and clarification questions in a satisfactory manner (Q2 - Q4). Please try to address them in a future manuscript version, some of these details are key to the method (e.g. trajectory stream attending to multiple past frames and L_refine definition etc) but are not sufficiently well described.
>
> Wrt my Q1 I am not particularly reassured yet. Please explain what your new pitch is in 2 sentences. Also, while in theory your method can benefit from longer historical information, in practice you have not tried it, so it is not proven. What seems proven to me is that presenting the information in a streaming fashion may be superior.
> Fig 1 is confusing because it shows that for standard methods, each frame attending only its own timestep independently. This is not correct since it attends X frames of previous data but it is not shown (yet the past frames are shown in yellow for RealMotion).

---

> ### Author Response · Authors · 2024-08-10
>
> Thanks for the recognition.
>
> We cannot provide the evaluation of longer history due to there is no temporal relationships between different scenes in existing benchmarks. Once such longer history data becomes available, it is definitely insightful to test our approach then. That being said, we have now tested the longest possible history with the existing benchmark sampled from the real-world driving data.
>
> Regarding Fig.1, please note each blue circle represents a specific scene rather than a single frame, which consists of historical trajectories and HP map within a certain range. In this case, current methods independently process every individual scene, while RealMotion is designed to model the temporal relationships across successive scenes. We will further clarify to minimize similar misunderstanding.

---

> > ### Comment · Reviewer_fgWb · 2024-08-10
> >
> > > We cannot provide the evaluation of longer history due to there is no temporal relationships between different scenes in existing benchmarks. Once such longer history data becomes available, it is definitely insightful to test our approach then.
> >
> > This is understandable. But this is an important point/disclaimer, and it is not actually mentioned in the intro etc.
> > I also had asked: "please explain what your updated intro contribution statement relative to existing methods is, in 2 sentences" to see how it would address my overclaiming concerns. I note that your response does not address this ask.
> >
> > > Regarding Fig.1, please note each blue circle represents a specific scene rather than a single frame.
> >
> > I suggest at the very least you augment the figure caption to clarify that each blue scene circle contains X seconds of past history, or some such.

---

> ### Author Response · Authors · 2024-08-11
>
> > please explain what your updated intro contribution statement relative to existing methods is, in 2 sentences" to see how it would address my overclaiming concerns.
>
> Thanks. Existing methods tackle motion forecasting task in a scene-independent manner, which could limit the model performance in real-world settings where motion is typically continuous while ego-car drives on. Upon this observation, we propose temporal relationship modeling across scenes and introduce an effective approach, RealMotion, to realize this idea in a two stream framework.
>
> > I suggest at the very least you augment the figure caption to clarify that each blue scene circle contains X seconds of past history, or some such.
>
> Thanks for the suggestion and we will revise as suggested.

---

> > ### Comment · Reviewer_fgWb · 2024-08-12
> >
> > Thank you for all the clarifications. Overall I am leaning to accept, as the ideas and results generally warrant it, and will maintain my score.
> >
> > In doing this, I assume you will temper claims that existing solutions are "unrealistic to use in practice" even if you seem to be able to improve on them, and will explicitly call out the fact that you have not actually validated benefits from being able to process longer histories, per se, which seems left for future work.

---

> > > ### Author Response · Authors · 2024-08-12
> > >
> > > We appreciate the reviewer's time for reviewing and thanks again for the valuable comments. We will revise and refine the paper as suggested in the revision.

---

> > > > ### Comment · Area_Chair_D1Jd · 2024-08-12
> > > >
> > > > I thank the reviewer for providing a thorough feedback and fully support the importance of appropriate positioning for this work. For a positive decision on the submission the authors will have to explicitly commit to comprehensively updating the abstract and introduction of the paper to accurately reflect the contributions (not just changing a few words here and there), as well as reporting SEPT results in the main paper (with a disclaimer that the setting are not fully comparable) and thoroughly discussing the missing related work (not just adding citations).
> > > >
> > > > Reviewer fgWb, please comment if the title of the paper is adequate or requires updating as well.

---

> > > > > ### Author Response · Authors · 2024-08-13
> > > > >
> > > > > Thanks. We will make sure to update abstract and introduction section to accurately reflect the contributions, report SEPT results, and thoroughly discuss the similarities and differences between missing related work and ours.

---

### Author Rebuttal · Authors · 2024-08-06

We provide additional figures in the PDF files for more intuitive demonstration.

---

### Decision · Program_Chairs · 2024-09-25

**Decision:**

Accept (spotlight)

**Comment:**

The paper received unanimous acceptance recommendations, with reviewers praising the novelty and importance of the problem studied, as well as the strong experimental results of the proposed approach. However, serious concerns were raised regarding overclaiming in the paper's positioning and the omission of related work. In their rebuttal and subsequent discussion, the authors committed to comprehensively updating the abstract and introduction to accurately reflect the paper’s contributions—specifically, by framing the contribution as "following a streaming paradigm and designing more domain-appropriate fusion mechanisms across time than the traditional vanilla MLP or transformer operations," as suggested by Reviewer fgWb. Additionally, the authors agreed to discuss and include the missing related works in the experimental evaluation.

After considering the paper, reviews, rebuttal, and further discussion, the area chair decided to accept the paper, conditional on the revisions detailed above. The area chair will review the camera-ready version to ensure that these conditions have been met.